# Exploring Matching Rates: From Key Point Selection to Camera Relocalization

## ABSTRACT

Camera relocalization is the task of estimating camera pose within a known scene. It has important applications in the fields of Virtual Reality (VR), Augmented Reality (AR), robotics, and more within the domain of computer vision. Learning-based camera relocalizers have demonstrated leading pose accuracy, yet all current methods invariably utilize all the information within an image for pose estimation. This may offer robustness under challenging viewpoints but impacts the localization accuracy for viewpoints that are easier to localize. In this paper, we propose a method to gauge the credibility of image pose, enabling our approach to achieve more accurate localization on keyframes. Additionally, we have devised a keypoint selection method predicated on matching rate. Furthermore, we have developed a keypoint evaluation technique based on reprojection error, which estimates the scene coordinates for points within the scene that truly warrant attention, thereby enhancing the localization performance for keyframes. We also introduce a gated camera pose estimation strategy, employing an updated keypoint-based network for keyframes with higher credibility and a more robust network for difficult viewpoints. By adopting an effective curriculum learning scheme, we have achieved higher accuracy within a training span of just 20 minutes. Our method's superior performance is validated through rigorous experimentation. The code will be released.

## CCS CONCEPTS

• **Computing methodologies** → **Perception**; *Mixed / augmented reality*; *Virtual reality*; *Machine learning*.

## KEYWORDS

camera relocalization, scene coordinates regression, keypoint sets, keypoint guided

## 1 INTRODUCTION

The concept of certainty is of profound importance in human endeavors, providing solace through the predictability of contemporary life. This pursuit of dependability permeates the field of camera relocalization, wherein there is a desire for pose estimation to be imbued with a similar degree of certainty. camera relocalization

*ACM MM, 2024, Melbourne, Australia*

© 2024 Copyright held by the owner/author(s). Publication rights licensed to ACM.
ACM ISBN 978-x-xxxx-xxxx-x/YY/MM
https://doi.org/10.1145/nnnnnnn.nnnnnnn

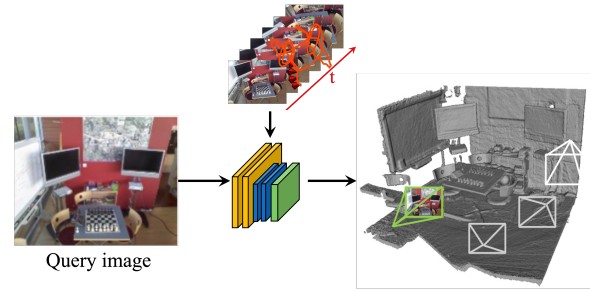

Query image

**Figure 1: Camera Relocalization. Camera relocalization is the task of estimating the camera pose within a known scene. Our approach endeavors to incorporate confidence measures and keypoint cues into the process of camera pose estimation, thereby providing credibility support during practical applications and further enhancing the positional accuracy of keyframes**

tasks necessitate the computation of a camera's 6-DoF pose relative to a pre-mapped scene when presented with query images. Training involves the ingestion of image sequences paired with their respective camera poses, with the aim of facilitating rapid and precise determination of the camera's position upon its subsequent scene encounter. This functionality is pivotal in diverse applications such as VR, AR, robotics, and autonomous navigation. Predominant research efforts in this area strive to refine the accuracy of camera pose estimations uniformly across all test images. Nonetheless, a nuanced analysis uncovers intrinsic limitations of this approach. Consequently, the ability to assess the likelihood of obtaining an accurate camera pose from an input image is equally critical as achieving high precision in the pose estimation itself.

camera relocalization's evolution commenced with image retrieval, the primitive form of which, scene recognition, depended on robust image retrieval for localization. Progressive iterations in camera relocalization have focused on feature-based methods to augment precision. The introduction of random regression forests [11, 24], capable of regressing scene coordinates, markedly enhanced camera relocalization accuracy. Deep learning frameworks [14–16], utilizing deep networks for direct pose regression from images, have validated the proficiency of these networks in correlating imagery with their respective poses, albeit with limitations in generalization across novel viewpoints.

Scene coordinate regression [2, 4–7] signifies a dissection of the camera relocalization task, assigning the network the exclusive function of correlating input imagery with scene coordinates. The ACE[2] framework advances this task decomposition, with a feature backbone isolating scene feature descriptors, and scene coordinate regression dedicated to aligning these descriptors with scene coordinates. This bifurcation lessens the computational demand on

scene-specific network models, thus expediting camera relocalization.

We advocate for a further subdivision of the camera relocalization task. Specifically, the identification of inliers should be decoupled from the RANSAC and PnP procedures. This can be realized by employing a convolutional network to assist in keypoint evaluation. In parallel, this network can be harnessed, in conjunction with inlier data from the estimated pose, to ascertain the confidence of the camera pose estimation.

To achieve our objectives, we have devised a pose estimation confidence assessment method based on the inlier rate, and on this foundation, we have developed a keypoint judgment network and training framework to further enhance the accuracy and assessment capabilities of camera relocalization. Specifically, our contributions can be summarized as follows:

- We have designed a scene coordinate estimation method and a camera pose estimation framework based on a keypoint network.

- We have developed a pose estimation confidence method based on the keypoint discrimination network.

- Through extensive experimentation, we have demonstrated that our method outperforms the state-of-the-art methods in accuracy and validated the effectiveness of each component of our approach.

## 2 RELATED WORK

The problem of camera relocalization involves recovering the 6DoF camera pose of a query image from a known scene. Current mainstream approaches include image or feature retrieval-based methods, pose regression methods, and scene coordinate regression methods. A more detailed introduction to these methods will be provided below.

The implementation of image retrieval methods involves searching a query image [21] from a database of images with poses to retrieve similar images. The retrieved image with the most similar pose is then outputted. Arandjelovic et al. proposed VLAD [26] for global image feature description, enabling the retrieval of similar images. NetVLAD [1] improved upon VLAD features by using a convolutional neural network model. It introduced a learnable generalized VLAD layer, enabling a plug-and-play approach for network-based feature extraction. InLoc [25] provided a dataset with panoramic images that had variations in lighting and dynamics. It proposed a method to optimize pose estimation using synthesized virtual views. By extracting and matching feature points [10], it established numerous matching relationships with noisy points. Robust pose estimation algorithms, such as RANSAC, significantly enhance the pose estimation accuracy. The main challenges of the above methods are the speed of database retrieval[1] and the accuracy of pose estimation. As the dataset size increases, the number of retrievals and computational complexity also increase.

Methods based on random forests [24] can be applied to feature extraction and matching [13]. They use a random forest composed of

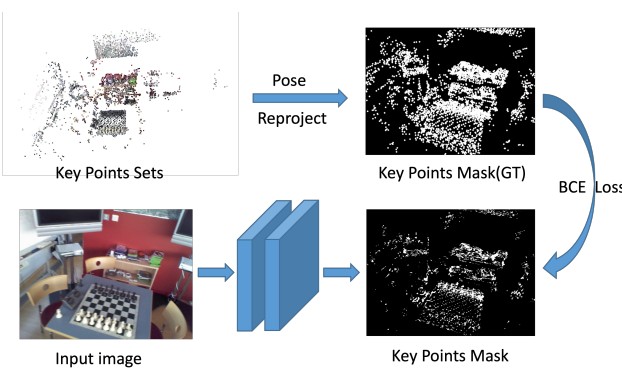

**Figure 2: Training the Keypoint Selection Network. When training the keypoint selection network, we obtain the mask of keypoints by projecting a set of keypoints. The keypoint selection network analyzes the features of the image to select those keypoints with a higher matching rate.**

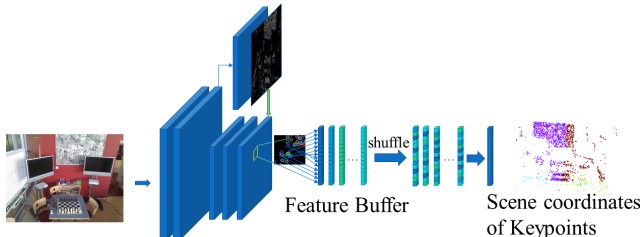

**Figure 3: Training Pipeline. For keypoint guided scene coordinate regression network training, we begin by extracting image features using a feature backbone. These features are then passed into a keypoint estimation network. Following this, features are filtered based on the estimated keypoints. The selected features, along with their corresponding parameters, are stored into a Training Buffer for the training process. Finally, the network is trained expeditiously using Gradient Decorrelation and Curriculum Learning techniques.**

regression decision trees to regress scene coordinates, which represent the coordinates of image pixels in the scene model. Method [9] based on random forests enables real-time learning of new scenes. Method [11] enhances camera relocalization in dynamic scenes with moving objects by suppressing the routing process based on the stability of features. However, random forest methods use depth information as input features.

The PoseNet series [14–16] utilizes convolutional neural networks to extract features from images and directly regress camera poses. In subsequent PoseNet methods, PoseNet16 [14] improved localization results by controlling the model's confidence through uncertainty measurement and determining the correlation between input images and the scene. PoseNet17 [15] explores effective training methods and focuses on designing loss functions for direct pose regression.

In DSAC [4], a probabilistic model derived from reinforcement learning makes the optimal selection of RANSAC differentiable. This

allows deep learning-based methods to be applied to camera relocalization in an end-to-end manner. DSAC++ [5] improves the network model, training methods, and data representation, further enhancing accuracy. ESAC [6] uses a hybrid expert model based on [4] to address the coverage of large datasets and ambiguity problems. DSAC* [7] improves [4] by using a better Resnet network structure and an improved training loss function, further enhancing pose estimation. AECRN [17] introduces RWEI to represent event data, enabling its effective application to scene coordinate regression. By designing an attention-based network architecture, it achieves higher accuracy in event-based scene coordinate regression. By extracting and matching feature points [10], it established numerous matching relationships with noisy points. ACE [2] further refines the DSAC series methods [4–7]by separating feature extraction from scene coordinate regression and using curriculum learning to reduce the training time of the scene coordinate regression head to around 5 minutes. This greatly reduces the deployment time of the model in new scenes. However, [2] neglects the estimation of important coordinate points. It estimates all points, neglecting both the estimation of image credibility and the consideration of high-confidence images and pixels. In this paper, we propose a method that measures the credibility of image pose estimation and guides scene coordinate estimation using key points, achieving further improvement in accuracy.

## 3 METHOD

The previous ACE method [2] proposed a novel concept of dividing scene coordinate regression into two parts: the first part involves extracting features from the image, and the second part focuses on regressing the scene coordinates from the extracted features. This raised another question: can we integrate keypoint guidance into this process to enhance the scene coordinate regression capability of the regression head for those points critical to accurate localization in real scenes, thus achieving more precise camera relocalization? To address this, we designed a keypoint discrimination network. Before training the scene coordinate regression head, we trained a keypoint discrimination network to determine whether a particular feature is a key point in the scene that is essential for camera relocalization or scene recognition. Utilizing these keypoints, we determine the features and points to be used for scene coordinate regression learning.

### 3.1 Identifying Key Points by Highest Matching Rate

The selection of key points is, in fact, a highly challenging endeavor, yet it is crucial for localization tasks. This is because localization relies on the establishment of matching relationships between images, and the foundation of these relationships is the pairing of key points. The selection and matching of key points also represent the primary method by which humans and all other visual animals perform localization. A common procedure involves initially identifying points that are easily recognizable within an observation, followed by continuous tracking of these points and estimation of one's own pose. Inspired by insights from SiLK [13], we posit that the fundamental criteria for key points should be their ease of matching and the likelihood of correct matches. Based on this, we define those key points through the highest matching rate.

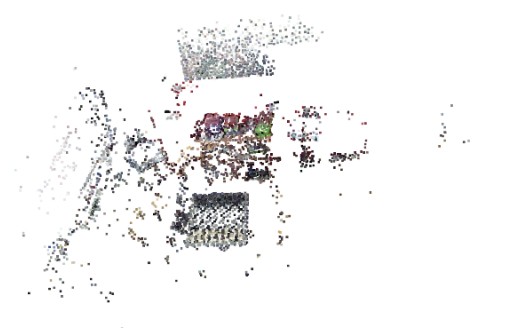

Figure 4: Selected Keypoints with the Highest Matching Rate in chess scene. These key points encompass nearly all regions of the scene that are readily observable while discarding areas with ambiguous interpretations.

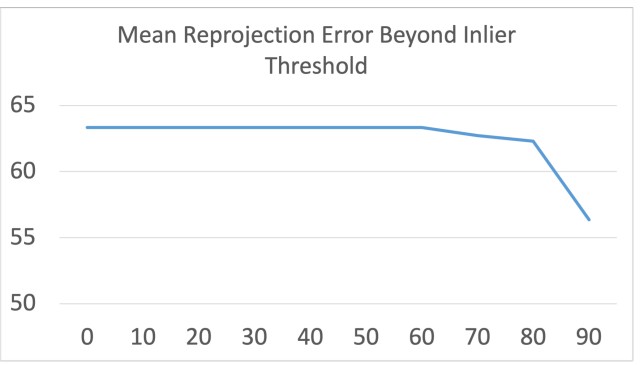

Figure 5: The blue curve represents the mean translation error beyond the Inlier ratio.

Let $P = \{p_1, p_2, \ldots, p_n\}$ be the set of all points in the image, and let $M(p)$ be a function that measures the matching rate of point $p$. We define the key points $K$ as follows:

$$K = \{p \in P \mid M(p) \geq \tau\}, \quad (1)$$

where $\tau$ is a threshold value chosen based on the desired confidence level for the matches. The function $M(p)$ is designed to reflect the ease of matching and correctness of the matches, which can be formulated as:

$$M(p) = \sum_{i=1}^{n} w_i \cdot \text{match\_quality}(p, p_i), \quad (2)$$

Here, $w_i$ are weights assigned based on the relative importance of matching with point $p_i$, and $\text{match\_quality}(p, p_i)$ is a function that returns a score representing the quality of the match between points $p$ and $p_i$.

$$\text{match\_quality}(p, p_i) = \exp\left(-\frac{\|\text{desc}(p) - \text{desc}(p_i)\|^2}{2\sigma^2}\right), \quad (3)$$

Figure 6: Pose Estimation Framework. Our pose estimation can be construed as performing pose selection across two distinct branches. The upper path is a keypoint-guided branch endowed with image confidence estimation, aimed at more accurately estimating the camera poses of images with elevated confidence levels. The lower path is a generalist branch tasked with estimating the camera poses of images with a higher degree of generalizability but without assured accuracy. Initially, query images undergo feature extraction via a Feature Backbone, and the resultant features are inputted into both a Keypoint Selector Network and a pair of Scene Coordinate Estimators. Subsequently, within the keypoint branch, only keypoints are leveraged for pose estimation, which in turn generates a confidence metric, specifically the Inlier Ratio. Ultimately, a gated mechanism in the Pose Selection phase adjudicates between the poses from the high-confidence branch or the more generalist branch, with the adjudicated pose being promulgated as the definitive pose output.

where $desc(p)$ represents the feature descriptor of point $p$, and $\sigma$ is a scaling parameter that adjusts the sensitivity of the match quality to differences in the descriptors.

We utilized Colmap [22] to obtain the collection of keypoints, upon which we performed projection and optimization. The figure 4 visualizes the finalized selection of our keypoint collection. It is evident that our keypoint collection encompasses the primary and pivotal regions of the scene while disregarding areas that are deficient in texture or lack identifiable qualities.

## 3.2 Further Optimization of Point Selection Using Reprojection Error

Selecting effective inliers in keypoint-guided inlier ratio estimation requires the network to determine which points should be considered as keypoints and which points generate more accurate 3D scene coordinates, leading to more correct 2D-3D correspondences. There are various criteria for determining keypoints. Traditional methods control the number of keypoints through non-maximum suppression, while some newer convolutional neural network-based methods use matching accuracy to evaluate keypoints. However, for convolutional networks and scenes, each model's keypoints in each scene actually exhibit randomness and uncertainty. Therefore, we propose a keypoint probability evaluation method based on reprojection error, converting the reprojection error into keypoint probabilities using the following formula:

$$k_i = 1 - \frac{1}{1 + e^{\lambda - p_i}}, \qquad (4)$$

where $\lambda$ is the softness parameter that controls the tolerance level of the reprojection error. A smaller value indicates that a smaller reprojection error is required to achieve a higher keypoint probability. In this work, we set $\lambda$ to 4. $p_i$ represents the reprojection error of the scene coordinate estimation, and $k_i$ represents the keypoint probability of that point. This formula is inspired by the sigmoid function. We further estimate the keypoint probabilities based on the trained scene coordinate estimation head. We found that after feature extraction by the convolutional network, the performance of the scene coordinate regression head in regressing scene coordinates exhibits randomness, meaning it does not always perform well at a specific point but rather shows a probabilistic behaviour. Therefore, we estimate the keypoint probabilities based on the trained scene coordinate estimation head, making our keypoint estimation method scene- and model-specific. This requires effective and efficient training of our network. Thus, we employ a simple network structure and accelerate the training of the keypoint selection network using curriculum learning. This approach enables us to achieve our goal of efficient training with minimal additional time.

## 3.3 Inlier Ratio Based Confidence Estimation

Our priority should be ensuring the quality of our outputs rather than maximizing the number of outputs. Currently, all vision-based camera relocalization methods aim to improve overall performance

**Table 1: Pose re-localization results compare with other methods.**

| Scene | DSAC | DSAC++ | Cas. (3D) | DSACStar | ACE | **Ours** |
|-------|------|--------|-----------|----------|-----|----------|
| chess | 94.60% | 93.80% | 99.95% | 96.70% | 100% | 100% |
| fire | 74.30% | 75.60% | 99.70% | 92.90% | 99.50% | 99.90% |
| heads | 71.70% | 18.40% | 100.00% | 98.20% | 99.70% | 100% |
| office | 71.20% | 75.40% | 99.48% | 87.10% | 100% | 99.50% |
| pumpkin | 53.60% | 55.90% | 90.85% | 60.70% | 99.90% | 99.90% |
| redkitchen | 51.20% | 50.70% | 90.68% | 65.30% | 98.20% | 99.70% |
| stairs | 4.50% | 2.00% | 94.20% | 64.10% | 81.90% | 88.60% |
| Average | 60.10% | 60.40% | 96.41% | 80.71% | 97.03% | 98.23% |

**Table 2: Pose re-localization results compare with other methods on Cambridge Landmarks dataset.**

| | | Cambridge Landmarks | | | | | Average |
|---|---|---|---|---|---|---|---|
| | | Court | King's | Hospital | Shop | St.Mary's | (cm/°) |
| FM | AS (SIFT) | 24/0.1 | 13/0.2 | 20/0.4 | 4/0.2 | 8/0.3 | 14/0.2 |
| | hLoc(SP+SG) | 16/0.1 | 12/0.2 | 15/0.3 | 4/0.2 | 7/0.2 | 11/0.2 |
| | pixLoc | 30/0.1 | 14/0.2 | 16/0.3 | 5/0.2 | 10/0.3 | 15/0.2 |
| | GoMatch | N/A | 25/0.6 | 283/8.1 | 48/4.8 | 335/9.9 | N/A |
| | HybridSC | N/A | 81/0.6 | 75/1.0 | 19/0.5 | 50/0.5 | N/A |
| APR | PoseNet17 | 683/3.5 | 88/1.0 | 320/3.3 | 88/3.8 | 157/3.3 | 267/3.0 |
| | MS-Transformer | N/A | 83/1.5 | 181/2.4 | 86/3.1 | 162/4.0 | N/A |
| SCR w/Depth | DSAC*(Full) | 49/0.3 | 15/0.3 | 21/0.4 | 5/0.3 | 13/0.4 | 21/0.3 |
| | SANet | 328/2.0 | 32/0.5 | 32/0.5 | 10/0.5 | 16/0.6 | 84/0.8 |
| | SRC | 81/0.5 | 39/0.7 | 38/0.5 | 19/1.0 | 31/1.0 | 42/0.7 |
| SCR | DSAC*(Full) | 34/0.2 | 18/0.3 | 21/0.4 | 5/0.3 | 15/0.6 | 19/0.4 |
| | DSAC*(Tiny) | 98/0.5 | 27/0.4 | 33/0.6 | 11/0.5 | 56/1.8 | 45/0.8 |
| | ACE | 43/0.2 | 28/0.4 | 31/0.6 | 5/0.3 | 18/0.6 | 25/0.46 |
| | Ours | 46/0.5 | 21/0.4 | 23/0.6 | 5/0.4 | 12/0.6 | 22/0.48 |

across the entire test set. However, it is not realistic to expect camera relocalization methods to perform efficiently and consistently in challenging scenarios with limited sampled data or abundant repetitive textures, considering that even humans exhibit limited generalization abilities in scenes with restricted input. This can be observed from the practical applications of existing methods in various scenes. For instance, in the 7scenes dataset, the "stairs" scene is notably more challenging for relocalization. Even a random image from a particular viewpoint in this scene could result in misjudgment.

An approach from the perspective of human localization suggests continuously increasing confidence in their observations. When humans reach a certain viewpoint during the localization process and have sufficient confidence, they consider that they have accurately determined their position in the scene. Based on this, we believe that it is also necessary to evaluate the confidence of the input images and their output poses. The output pose of an image with high confidence can be considered a valid camera pose.

To address this, we propose an image pose confidence estimation based on the inlier ratio. The confidence estimation of an image is calculated using the following formula:

$$r_i = \frac{100 \cdot N^i}{N^a}, \quad (5)$$

where $r_i$ represents the pose estimation confidence of image $i$, $N^i$ is the number of inliers output by the RANSAC-based PnP algorithm, and $N^a$ is the total number of 2D-3D correspondences provided as input. When using this confidence measure for pose estimation, specific thresholds need to be set to determine the level of confidence. We consider confidence levels above 90% to be considerably confident, above 80% to be highly confident, above 60% to be moderately confident, and below 60% to be questionable. These threshold settings are based on our observations of the experimental data.

### 3.4 Key Point Guided Inlier Ratio Estimation

When evaluating the confidence of image pose estimation using the inlier ratio, we found that existing methods perform poorly on the test set, with a negligible proportion of considerably confident estimates in the "stairs" scene. This is because current methods use all constructed 2D-3D correspondences for pose estimation indiscriminately with the selection of inliers solely determined

by RANSAC. To improve the inlier ratio, we propose keypoint-guided inlier ratio estimation. Our approach is based on the idea that we should assess the suitability of 2D-3D correspondences for scene coordinate estimation during their construction. Therefore, we introduce an additional network branch in the relocalization framework to estimate the confidence of the scene coordinates of the network's estimated pixels. Utilizing a keypoint estimation network to estimate the probability of regressing correct inliers for each pixel, we decide whether to use that point for camera pose estimation. Consequently, our updated confidence estimation is given by:

$$r_i = \frac{100 \cdot N^i}{N^{ar}}, \qquad (6)$$

where $N^{ar}$ denotes the subset of 2D-3D correspondences with reliable confidence, as estimated using the keypoint network.

## 3.5 Network Training

In our study, we are required to train a total of four networks: one to obtain keypoints with the highest matching rate, one to acquire keypoints with the minimal reprojection error, and two additional networks dedicated to the initialization and further refinement of these two types of keypoints, respectively. Among all training processes, the network trained to identify keypoints based on the matching rate is the most time-consuming. This is attributed to the fact that the majority of points in an image are not keypoints, precluding targeted training. Conversely, all other training activities are confined to the regions of keypoints identified by the matching rate.

We have ascertained the ground truth probabilities for keypoints at each point, denoted as $GT$. Consequently, within our keypoint estimation network, we are fundamentally estimating a probability mask for the extraction of keypoints. Accordingly, we employ the Binary Cross Entropy (BCE) loss as the loss function for our keypoint estimation network. This approach is applied to both the keypoint selection of the Highest Matching Rate and reprojection error:

$$L_k(k, \hat{k}) = -\frac{1}{N} \sum_{i=1}^{N} \left( k_i \log(\hat{k}_i) + (1 - k_i) \log(1 - \hat{k}_i) \right), \qquad (7)$$

Here, $N$ is the number of samples, $k_i$ represents the ground truth keypoint probability for the $i$-th feature, and $\hat{k}_i$ represents the predicted keypoint probability for the $i$-th feature.

Consequently, during the supervision process, we are limited to methods such as re-projection error for guiding the learning of scene coordinates. Specifically, the loss used for supervising is:

$$\ell_\pi[\mathbf{x}_i, \mathbf{y}_i, \mathbf{h}_i^*] = \begin{cases} e_\pi(\mathbf{x}_i, \mathbf{y}_i, \mathbf{h}_i^*) & \text{if } \mathbf{y}_i \in \mathcal{V} \\ ||\mathbf{y}_i - \bar{\mathbf{y}}_i||_0 & \text{otherwise.} \end{cases}, \qquad (8)$$

where $\mathbf{x}$ is the 2D coordinate, $\mathbf{y}$ is the 3D scene coordinate, $\bar{\mathbf{y}}_i$ is the GT 3D scene coordinate, $\mathbf{h}^*$ is the GT pose, and $\mathcal{V}$ is the group of 3D scene coodinate satisfied with the determination of re-projection error $e_\pi$ in [2].

We have designed a novel scene coordinate estimation head to accommodate more complex scene conditions. Enhancing the network depth of the scene coordinate estimation head boosts its ability to solve scene coordinates in challenging scenarios. We follow the Curriculum Training technique adopted in [2] for training our scene coordinate regression network.

## 4 EXPERIMENTS

Our method was implemented using PyTorch, building upon the publicly available code from ACE [2]. Here we detail our primary parameter settings. For the training of the matching-based keypoint network, a ResNet-like architecture was adopted. The initial learning rate was established at 0.0001, and the AdamW optimizer was consistently employed across all network configurations. When initializing the typical scene coordinate regression head, we set up a training buffer of 8.8 million samples, with all features randomly sampled from random images and augmentations. For training the keypoint selection network, an 8 million sample training buffer was established, drawing features from randomly chosen images. It is noteworthy that we utilize all outputs from the sampled images for keypoint training. In training the keypoint-guided scene coordinate regression head, we fine-tuned the initialized head rather than starting the training anew, thus preserving the network's capability to regress scene coordinates for non-keypoint features. For the pose estimator, we continued to employ the method from ACE [2], but with improvements made to the code to accommodate our approach. When employing the keypoint estimation network as a guide, we select points with an estimated probability value greater than 0.8 as keypoints, corresponding to a reprojection error of approximately 5.

## 4.1 Indoor Relocalization

We conducted experiments on the 7Scenes [24] dataset, which offers a variety of small-scale indoor scenes captured with handheld devices, along with depth information and camera poses. We utilized the camera poses obtained via Structure from Motion (SfM) provided in [3] as ground truth (GT), as it has been validated in [3] that scene coordinate regression methods are more suited to pose estimation frameworks that employ a point matching mechanism akin to SfM. Our primary method of comparison was ACE [2], yet we also compared against other scene coordinate regression approaches, including precursors to ACE and methods based on random forests. In Table 1, we present the percentage performance of our method and others on the 7Scenes [24] dataset within 5 degrees and 5 centimeters of error. Our method is shown to further enhance relocalization performance, especially in challenging scenes such as 'stairs'.

During our experimentation, we found that our method not only excels in percentage performance within the $(5°, 5cm)$ error margin but also significantly improves the precision of pose estimation. We have detailed the performance of our method compared to others within $(2°, 2cm)$ and $(1°, 1cm)$ error margins in Table 4, respectively. The data reveal that the precision of camera pose estimates from our method has been further elevated, offering a solid guarantee for the practical application of camera relocalization technology in real-world scenarios.

**Table 3: Survival Rate and Accuracy.** We measure the survival rate across three different stages, defined as the ratio of pose estimates that meet our set confidence criteria, in conjunction with the accuracy of the pose estimation. We have set the confidence threshold to an inlier rate exceeding 90%. "Init." denotes the network post-initialization, utilizing all coordinate estimates for pose estimation; "Init. With K.S." refers to the post-initialization network employing keypoint selection for pose estimation; "Keypoint fine-tune with K.S." indicates the network after keypoint fine-tuning, using keypoint selection for pose estimation. Almost all pose estimates deemed to meet the confidence criteria fall within (5°,5cm), hence we only list the percentages for errors under (2°,2cm) and (1°,1cm).

| | Init. | | | Init. with K.S. | | | Keypoint fine tune with K.S. | | |
|---|---|---|---|---|---|---|---|---|---|
| | Survival Rate | 2°2cm | 1°1cm | Survival Rate | 2°2cm | 1°1cm | Survival Rate | 2°2cm | 1°1cm |
| chess | 97\|2000 | 100.00% | 91.80% | 1714\|2000 | 99.80% | 93.60% | 1718\|2000 | 99.80% | 93.90% |
| fire | 92\|2000 | 100.00% | 100.00% | 733\|2000 | 93.70% | 81.20% | 736\|2000 | 95.40% | 83.40% |
| heads | 0\|1000 | 0.00% | 0.00% | 238\|1000 | 100.00% | 98.70% | 236\|1000 | 100.00% | 99.20% |
| office | 20\|4000 | 100.00% | 25.00% | 3193\|4000 | 86.10% | 35.10% | 3197\|4000 | 87.40% | 42.50% |
| pumpkin | 48\|2000 | 100.00% | 22.90% | 1393\|2000 | 94.50% | 39.80% | 1388\|2000 | 97.60% | 54.70% |
| redkitchen | 38\|5000 | 100.00% | 100.00% | 3501\|5000 | 94.60% | 50.20% | 3483\|5000 | 96.40% | 63.60% |
| stairs | 0\|1000 | 0.00% | 0.00% | 9\|1000 | 100.00% | 0.00% | 29\|1000 | 100.00% | 24.10% |
| **Average** | 42\|2429 | 71.43% | 48.53% | 1540\|2429 | 95.53% | 56.94% | 1541\|2429 | 96.66% | 65.91% |

**Table 4: Results under Smaller Thresholds.** We further detail the camera relocalization accuracy for errors within (2°, 2cm) and (1°, 1cm). This improved precision can be attributed to the network's more meticulous learning of salient keypoints within the scene, which enables a deeper exploration of the scene's structure. Consequently, our method exhibits enhanced accuracy even within these reduced error margins.

| | Within (2°,2cm) | | | Within (1°,1cm) | | |
|---|---|---|---|---|---|---|
| | DSAC* | ACE | Ours | DSAC* | ACE | Ours |
| chess | 32.80% | 99.00% | 99.60% | 0.50% | 81.90% | 91.80% |
| fire | 55.20% | 87.10% | 95.20% | 14.80% | 57.00% | 63.90% |
| heads | 87.30% | 98.20% | 98.90% | 40.00% | 85.30% | 87.10% |
| office | 32.10% | 81.00% | 91.20% | 5.90% | 28.00% | 60.40% |
| pumpkin | 19.80% | 84.50% | 88.90% | 4.70% | 27.00% | 60.7% |
| redkitchen | 14.90% | 87.00% | 94.90% | 2.60% | 45.50% | 73.90% |
| stairs | 11.40% | 24.10% | 38.00% | 1.10% | 4.00% | 8.10% |
| **Average** | 36.21% | 80.13% | 86.67% | 9.94% | 46.96% | 63.70% |

## 4.2 Outdoor Relocalization

We validated our method for outdoor relocalization on the Cambridge Landmarks [16] with the results tabulated in Tables 2. For comparison, we replicate experiments from prior research using various methods: AS (SIFT) [20], hLoc(SP+SG) [18], pixLoc [19], Go-Match [28], HybridSC [8] for FM; PoseNet17 [15], MS-Transformer [23] for APR; DSAC*(Full) [7], SANet [27], SRC [12] for SCR w/Depth; DSAC*(Full) [7], DSAC*(Tiny) [7], ACE [2] for SCR. The Cambridge Landmarks dataset consists of images of various historical buildings in the old town area of Cambridge, with ground truth poses obtained using the SfM [3]. Our method performed exceptionally well in some scenes, outperforming our main comparison method, ACE, in most scenarios. This is attributed to our network's enhanced capability for scene coordinate inference. During our experiments, we also observed that a significant reason for poorer performance on outdoor data was the sparsity of the dataset. Our method showed commendable performance on the training set, yet its capacity to generalize to new viewpoints was somewhat limited. Overall, our

method's performance on outdoor data was superior to our main comparison method, ACE [2].

## 4.3 Ablation Study

*4.3.1 The Relationship Between Inlier Ratio and Error.* Our method introduces a confidence-based scene coordinate estimation approach. We ponder the potential outcomes if we solely rely on the poses of images with high confidence. We have detailed the performance of our method across various scenes at different confidence levels in the tables. To validate the appropriateness of our confidence measure, we plotted a curve representing the relationship between the confidence utilized and the translational error, as illustrated in Figure 3. For clarity, we scaled and averaged the data. The blue curve in the figure represents confidence, while the orange curve indicates the translational error. It is observable from the graph that an increase in error tends to follow a decrease in confidence. Although this correlation is not absolute, our observations suggest that there is generally an inverse relationship between inlier rate

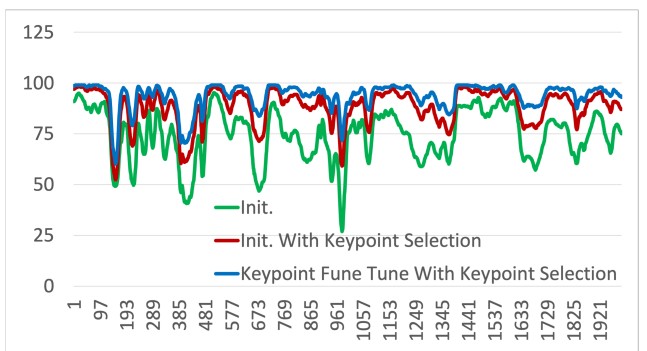

**Figure 7: Variations in the inlier ratio across different stages. We present the inlier rate progression over three distinct phases on the chess test dataset. The green curve represents the inlier rate following initialization. The orange curve depicts the inlier rate after the keypoint estimation network has been trained and directly applied to the initialization network.**

and error, implying that a higher inlier rate may be associated with increased error.

*4.3.2 Confidence Frame and Key Point Guided.* Enhancement of Confidence in Keypoint-Guided Networks. Our method aims to further improve the accuracy of images with high confidence; hence, we estimate scene coordinates on keypoint data and employ the inlier rate as a measure of confidence for the precision of image pose estimation. We preserve images with a confidence level of 90% and estimate their camera poses. Table 3 lists the survival rate and accuracy of our method at various stages for test images. It is evident that the survival rate of test images was quite low initially without keypoint guidance. The introduction of keypoint guidance resulted in a significant increase in the survival rate, where survival rate refers to images we consider to have sufficiently high accuracy. We also observe that the survival rate of the network, further optimized using keypoint guidance, did not decrease significantly, while the localization accuracy for the test data with high survival rates was further improved within error margins of $(2°, 2cm)$ and $(1°, 1cm)$. Moreover, within the subset of high-confidence images, we nearly achieved 100% accuracy within an error range of $(5°, 5cm)$, which validates the practical reliability of our method. Additionally, we note that in the Stairs scene, our confidence measurement method was not able to maintain a high survival rate. This could be attributed to the scene's higher difficulty and the presence of more repetitive textures, a phenomenon also observed in the Heads scene.

*4.3.3 Usage of Time and Computational.* Our approach does not significantly increase training duration or computational resource usage under standard conditions. The total training time is approximately around 20 minutes in two Nvidia GeForce 2080 Ti GPUs, which is acceptable considering the desired higher accuracy. During testing, when the survival rate meets our satisfaction, our computational load is lower than that of ACE. This is because we use fewer but more accurate 2D-3D matching points, reducing the required iterations. However, if the survival rate is suboptimal, we may incur

the cost of performing pose estimation twice: first, a confidence-based pose estimation, and then a more generalized pose estimation. When pose estimation is solely based on confidence, we use fewer 2D-3D correspondences for pose estimation.

*4.3.4 Limitations.* We found several shortcomings that require further research and improvement. Firstly, our method faces the challenge of a low number of keypoints in difficult scenes with sparse textures. Consequently, fine-tuning the process becomes nearly infeasible, and we heavily rely on the initialized generalized network. Secondly, our approach is limited by the confidence threshold. Currently, we use a fixed confidence threshold, but we have noticed that its performance varies across different scenes.

## 5 CONCLUSION

Measuring the confidence of pose estimation is critical in practical applications. In this paper, we present a method for assessing the credibility of image pose estimates, enabling effective evaluation of camera poses after estimation. Additionally, to improve the effective assessment of confidence and enhance localization accuracy, we have designed a keypoint evaluation method based on reprojection error. This method estimates scene coordinates for keypoints of interest, optimizing keyframe localization performance. Furthermore, in practical application, our designed gated camera pose estimation strategy, based on confidence thresholding, combines keypoint-guided networks with more generalized networks to further enhance the camera relocalization accuracy. Notably, our approach does not significantly increase training duration or volume compared to state-of-the-art methods, achieving greater accuracy within a training period of just 20 minutes. Through extensive experimental comparisons, we have demonstrated the effectiveness of our proposed method, surpassing state-of-the-art results.

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
