# OpenReview forum: "Exploring Matching Rates: From Key Point Selection to Camera Relocalization"
_acmmm.org/ACMMM/2024/Conference — MM2024 Oral_

### Official Review · Reviewer_EvXe · 2024-05-23

**Rating:** 4
**Confidence:** 3

**Summary:**

This paper addresses the camera relocalization task, which estimates the camera pose of a query image in a known 3D scene. The proposed method extends the ACE framework to two branches and introduces a keypoint network, pose confidence estimation, and gated mechanism in the pose selection. The proposed method identifies keypoints by the highest matching rate, converts reprojection errors into keypoint probabilities, and trains the keypoint network using binary cross-entropy loss based on the keypoint probabilities. Evaluation experiments using the 7 Scenes and Cambridge Landmarks datasets demonstrate the effectiveness of the proposed method.

**Strengths:**

1. The idea of improving the relocalization accuracy by selecting only the features essential for camera relocalization is interesting.
2. The modules are reasonably motivated and designed, and the proposed method is technically sound.
3. The proposed method showed improvement in relocalization accuracy in most scenarios compared to the state-of-the-art method ACE.

**Limitations:**

1. This paper claims to propose a method of measuring pose estimation confidence based on inlier ratio. However, measuring pose confidence based on the inlier ratio is part of the basic ideas of the RANSAC family and is not a novel idea.
2. The algorithm of the proposed method has some unclear parts. From the description in section 4.3.3, it can be interpreted that generalized pose estimation is performed only when the survival rate is suboptimal. However, from Figure 6 and its caption, it can be interpreted that both confidence-based pose estimation and generalized pose estimation are always performed. These seem contradictory, and it is also unclear what suboptimal represents. The authors should clearly describe these.
3. Experiments lack a comparison with ACE Ensemble [2]. Since the proposed method increases training time to 20 minutes, a comparison with this method should be considered. If this method is included in the comparison, the proposed method may not have state-of-the-art performance.
4. Experiments lack sufficient evaluation of the identifying key points proposed in section 3.1. The manuscript does not contain a specific value for the threshold τ. How does changing this threshold affect the re-localization results?
5. The effectiveness of the proposed gating mechanism in pose selection phase has not been demonstrated. Can poses with higher estimated confidence consistently achieve more accurate re-localization results?
6. In Tables 1, 3, and 4, the two decimal places are zero in most cases, which is unnatural. The authors should explain why such results were obtained.
7. Figures 1, 2, and 5-7 are not referenced from the text. Also, the authors refer to Figure 3 on line 806, but Figure 7 is probably correct. Furthermore, Table 4 is referenced before Tables 2 and 3. Figures and tables should be appropriately referenced from the text.
8. There are several format errors in the manuscript.  For example, the initial letter should be capitalized in line 97, and a space is lacked before the word "by" in  line 246. There are also some typos, e.g., "coodinate" on line 632.
9. The manuscript contains many spelling inconsistencies. For example, a keypoint network is sometimes referred to as a keypoint judgment network, keypoint discrimination network, keypoint selection network, keypoint estimation network, keypoint classification network, or keypoint selector, which can be confusing.

**Suitability:**

2

---

### Official Review · Reviewer_hccy · 2024-05-24

**Rating:** 3
**Confidence:** 3

**Summary:**

This paper presents a straightforward and effective method to enhance the accuracy of camera pose estimation by improving the correctness of keypoint and keyframe matches.

**Strengths:**

1. The authors propose a method to evaluate image confidence, analyzing each image's likelihood of yielding a valid pose, thereby enhancing the accuracy of the captured information.
2. By leveraging different networks, the authors obtain reliable keypoints contributing to improved pose estimation precision.
3. Ablation experiments are conducted to validate the effectiveness of individual networks, demonstrating slight improvements over current state-of-the-art approaches.

**Limitations:**

1. Although comparisons are made with the most advanced existing methods, the limited number of comparative methods in the experiments is insufficient to prove the superiority of this proposed method over others. For instance, SANet[1] and SRC[2].
[1] Yang, Luwei, et al. "Sanet: Scene agnostic network for camera localization." IEEE International Conference on Computer Vision 2019.
[2] Dong, Siyan, et al. "Visual localization via few-shot scene region classification." International Conference on 3D Vision. IEEE, 2022.
2. Both the methods for assessing image confidence and the confidence in keypoint matches rely on threshold-based determinations, which lack generalizability and may not be adaptable to different scenarios.
3. Scenes with low texture or repetitive patterns notably affect keypoint matching. Although the authors address the scenario of stairs in repetitive textures, they overlook discussions on other instances of repetitive or low-texture contexts, such as walls and railings, highlighting the need for more comprehensive experimental evaluations.
4. The subheading in Section 4.3 should be capitalized for consistency.
5. It would benefit the authors to elaborate on how the ground-truth probabilities for each key point in the dataset are obtained.

**Suitability:**

3

---

### Official Review · Reviewer_8Agz · 2024-06-03

**Rating:** 3
**Confidence:** 3

**Summary:**

The paper deals with the problem of camera relocalization via scene coordinate regression, followed by a PnP algorithm. The paper more specifically focuses on the keypoint selection process for the PnP algorithm. The proposed approach uses several ratios and scores to select the best keypoints most suited for localization. The paper advocates that more than the best scene representation, in the case of relocalization, the most suited keypoints should be what the network focuses on.

**Strengths:**

First of all, I find that the paper’s initial position is sound and should definitely be investigated. Focusing on a scene representation of suitable keypoints for pose estimation rather than global and extensive scene representation appears to be the way to go. Thus, the idea is interesting and the paper has merits as the evaluation tends to show, especially when showing improvement in localization even with a scene representation less suited for reconstruction (shown in supp. mat).

The proposed implementation looks interesting. Indeed, exploiting the scene coordinate regression scheme both in a “global” and “local” approach shows similarities with global/local features joint use in image retrieval, which showed great promise. The proposed ratios are simple but coherent and the results seem to prove it. I also appreciated the “survival rate” evaluation. Indeed, being able to have an a priori confidence estimation on the final pose has multiple practical applications and should be highlighted.

**Limitations:**

Although the paper has merits and I find both the idea and its implementation interesting, I feel that the main problem of the paper is the paper itself. I found it very hard to understand the ins and outs of the proposed method. First, the structure of the paper and the logical progress when detailing the approach were lacking to my opinion. Second, some aspects in the way things were explained limited an easy understanding of the method and are very detrimental to the paper. I will go into more details below but bear in mind that although some answers to the questions may seem evident, in my opinion, the reason they arise is due to the lack of clarity in the paper.

In my opinion, a typical example of the lack of clarity is the third sentence of the Introduction. Although the problem of camera relocalization is well defined in the first sentence of the related work, in the introduction, a convoluted and unclear formulation is used. Furthermore this sentence shows the need for a thorough pass to check for typos and especially missing capital letters.
Staying with easy to fix issues, some small aspects should be addressed :
the query image used in the figures is not always the same (it differs in Figure 2). It could be interesting to always have the same, especially when compared to Figure 4 and the set of selected keypoints that does not show the same area of the scene,
the figures are almost never referenced in the text, except for Figure 4. This, alongside the fact that they are placed very far from the (assumed) corresponding text is quite confusing,
some figures seem to not be present (to which Figure and curves is section 4.3.1 referring to ?),
the figure 6 comes maybe a bit late in my opinion. It could be further used to explain the process and limit the confusion in the method description,
in the tables, the colors red and blue are not explained. Although it seems to represent the first and second results, those are usually highlighted with bold and underlined. It is necessary to define the color code,
keypoints is used in an undiscriminated way for 2D and 3D points and that can become confusing.

I feel that the confused aspect of the paper is also reflected by the related work section. The several parts of the section are not linked together and there is no global overview of where the proposed approach is set with regards to related approaches. Furthermore, the state of the art on retrieval-based methods is quite outdated. Only regarding the features used for retrieval, many more recent methods exist, such as this one example (Zhu, Sijie, et al. "R2former: Unified retrieval and reranking transformer for place recognition." Proceedings of the IEEE/CVF Conference on Computer Vision and Pattern Recognition. 2023.) that could lead the authors towards a more recent related work. Furthermore, RANSAC is NOT a pose estimation algorithm. It can be used alongside one to make it more robust but that is it.

On the proposed approach, if I understood correctly, the keypoint selection network uses a 3D scene reconstruction from Colmap to compare the 2D keypoints extracted in the image with supposedly important points in the scene. However, are we sure that the points suited for and kept by the SfM reconstruction are the most-suited for camera relocalization. This is assumed to be true in the method but is there any more justification ? Furthermore, does it mean that to retrain the model on another scene, a novel 3D reconstruction of the scene must be performed or is the training of the keypoint selector scene-agnostic ?

To my opinion, the approach has merits but I feel that both a thorough refactoring and a step back to have more hindsights would greatly benefit the paper. Most importantly, I believe that introductory paragraphs outlining the reasoning (at different levels) in each part would help readers to understand the whole paper.

**Suitability:**

2

---

### Meta-Review · Area_Chair_tXwY · 2024-07-03

**Recommendation:** Accept (Oral)
**Confidence:** 3

**Metareview:**

The paper deals with the problem of camera relocalization through scene coordinate regression, followed by a PnP algorithm for which a keypoint selection process is specifically designed. The proposal is technically sound and the comparative evaluations (enriched with the ones provided in the rebuttal) show that the proposal consistently outperforms existing state-of-the-art competitors.

Shared by the 3 reviewers, the main weaknesses of the article rely on the clarity of the presentation for which parts need to be rewritten (also because of many spelling inconsistencies) by considering the reviewers recommendations, and on the unimodal data and application aspect which is not at the heart of the ACM MM community's priorities. Depending on the acceptance rate at the conference, this may call into question its acceptance for oral presentation, and tip him towards a poster presentation.